# Is the bispectral index monitoring protective against postoperative cognitive decline? A systematic review with meta-analysis

Timea Bocskai[1], Márton Kovács[2], Zsolt Szakács[3,4], Noémi Gede[3], Péter Hegyi[3,4,5,6], Gábor Varga[7], István Pap[2], István Tóth[2], Péter Révész[2], István Szanyi[2], Adrienne Németh[2], Imre Gerlinger[2], Kázmér Karádi[8☯], László Lujber[2☯]*

1 Department of Anaesthesiology and Intensive Care, Medical School, University of Pécs, Pécs, Hungary, 2 Department of Otorhinolaryngology (ENT), Medical School, University of Pécs, Pécs, Hungary, 3 Institute for Translational Medicine, Medical School, University of Pécs, Pécs, Hungary, 4 János Szentágothai Research Center, University of Pécs, Pécs, Hungary, 5 Division of Gastroenterology, First Department of Medicine, Medical School, University of Pécs, Pécs, Hungary, 6 Hungarian Academy of Sciences–University of Szeged, Momentum Gastroenterology Multidisciplinary Research Group, Szeged, Hungary, 7 Department of Oral Biology, Semmelweis University, Budapest, Hungary, 8 Institute of Behavioral Sciences, Medical School, University of Pécs, Pécs, Hungary

☯ These authors contributed equally to this work.
* lujber@gmail.com

**Data Availability Statement:** All relevant data are within the manuscript and its Supporting Information files.

## Abstract

### Background

Several studies have suggested that monitoring the depth of anaesthesia might prevent the development of postoperative cognitive decline. We aimed to conduct a meta-analysis to investigate the effects of bispectral index (BIS) monitoring in anaesthesia.

### Methods

We searched in six major electronic databases. Trials were included if they discussed anaesthesia with and without BIS monitoring or low (<50) and high ($\geq$50) BIS levels and which measured the risk of postoperative delirium (POD) and/or postoperative cognitive dysfunction (POCD).

### Results

We included fourteen studies in the systematic review, eight of which were eligible for meta-analysis. BIS proved to be protective against POD at 1 day postoperatively in a cohort of 2138 patients (16.1% vs. 22.8% for BIS vs. no BIS groups, respectively; relative risk [RR] 0.71; 95% confidence interval [CI] 0.59 to 0.85, without significant between-study heterogeneity $I^2 = 0.0\%$, $P = 0.590$). The use of BIS was neutral for POCD at 1 week but protective for POCD at 12 weeks (15.8% vs. 18.8% for BIS vs. no BIS groups, respectively; RR = 0.84, CI: 0.66 to 1.08), without significant between-study heterogeneity ($I^2 = 25.8\%$, $P = 0.260$). The neutral association at 1 week proved to be underpowered with trial sequential analysis. In the comparison of low BIS versus high BIS, the incidence of POD at 1 day was similar in the groups.

**Funding:** This study was supported by an Economic Development and Innovation Operative Program Grant (GINOP 2.3.2-15-2016-00048) and an Institutional Developments for Enhancing Intelligent Specialization Grant (EFOP-3.6.2-16-2017-0006) from the National Research, Development and Innovation Office. The funders had no role in study design, data collection and analysis, decision to publish, or preparation of the manuscript.

**Competing interests:** The authors have declared that no competing interests exist.

## Conclusion

Our findings suggest a protective effect of BIS compared to not using BIS regarding the incidence of POD at 1 day and POCD at 12 weeks. However, limitations of the evidence warrant further investigation to identify those groups of patients by age, comorbid conditions and other individual variables who would benefit the most from the use of BIS monitoring.

## Introduction

The disturbance in cognitive brain activity after surgery under general anaesthesia is worrisome. According to the literature, the incidence of cognitive decline after minor or major surgeries ranges from 7 to 29% in the elderly and even reaches 19% in younger patients as well [1–4]. A prominently high incidence was observed after heart surgery (80%) [5].

The aetiology of postoperative cognitive decline is multifactorial [1–4, 6–9]. Considering the causes of postoperative delirium (POD) and postoperative cognitive dysfunction (POCD), the predisposing factors are very similar [1, 2, 6, 7]. However, important differences may be found in the pathophysiological background [2, 6]. By definition, POD is an acute deterioration in cognitive function, a disturbance of consciousness. It develops in the early days postoperatively [2, 6, 7, 10, 11]. POCD is one of the adverse effects of anaesthesia which develops in the later postoperative period and manifests as a decline in a patient's cognitive abilities [2, 6, 7, 10]. Several neuropsychological tests are available to detect changes of cognitive function, although proper test assessment and interpretation are often problematic [2, 8].

Prevention of postoperative cognitive disturbances is a top priority, with one potential tool being the application of bispectral index (BIS) monitoring [2, 6, 7, 9, 12–14]. Previous systematic reviews have yielded discrepant results, although the methods applied in data collection, selection and pooling have been varied and, sometimes, incomplete as well [15–19]. These reviews have left the question open on how BIS monitor-guided anaesthesia influences the incidence of POD and POCD.

Our aim was to compile all available evidence on the effects of BIS on POD and POCD in a systematic review with meta-analysis using two comparisons: general anaesthesia with or without BIS monitoring and anaesthesia with low (BIS values <50) or high (BIS values ≥50) BIS levels.

## Materials and methods

Our study is a systematic review with a meta-analysis of randomized controlled trials (RCTs) evaluating the effect of BIS monitoring on patients under general anaesthesia in the prevention of POD and POCD. Our publication adheres to the PRISMA Statement [20].

### PICOS and eligibility

The review question was formulated by the PICO framework. We included studies that discuss (Population) adult patients who underwent general anaesthesia (Intervention$_1$ vs. Comparator$_1$) with BIS monitoring vs. without BIS monitoring or (Intervention$_2$ vs. Comparator$_2$) with low vs. high BIS and which measure (Outcome) the risk of POD and/or POCD. (Study design) As regards study design, we included RCTs exclusively.

A BIS value is a number on a spectrum between 0 and 100 without dimension, scaled to correlate with important clinical endpoints and electroencephalographic (EEG) signals under

anaesthesia [7, 12–14]. The upper end of the spectrum is the awake state with a typical BIS value near 100 [7, 12]. The lower end (BIS = 0) is defined as an isoelectric EEG record [7, 12]. The optimal BIS range of standard surgical anaesthesia falls between 40 and 60 [7, 12–14]. This range can be divided into low (<50) and high (≥50) BIS levels.

We defined POD as a complete disturbance affecting the integrity of consciousness in the first 1–5 days after surgery, whereas POCD begins days later from 1 week on and may persist for 4–6 weeks or even longer, up to 52 weeks [2–4, 10]. Our primary outcome was POD at 1 day, while secondary outcomes included POD at 2, 3, 4 and 5 days as well as POCD at 1, 12 and 52 weeks.

## Search, selection and data extraction

We searched electronic databases including MEDLINE (via PubMed), EMBASE, Cochrane Controlled Register of Trials (CENTRAL), SCOPUS, WHO Global Health Library/Global Index Medicus and Clinical Trial.gov for relevant articles from inception up to 29 April 2019. Human and English-language filters were imposed on the search, where appropriate. Further details of search, selection and data extraction are shown in S1 Appendix. We did not contact the original authors for further information.

## Risk of bias (RoB) assessment

We used the Cochrane Risk of Bias Tool to rate risk of bias along critical points in methodology (PH, GV, IP and IT) [21].

Results from the RoB assessment were incorporated into the interpretation of findings but not in statistical analysis (KK). Discrepancies during the assessment were resolved by reaching a consensus.

## Quality of evidence

We used the Grading of Recommendations, Assessment, Development and Evaluation (GRADE) approach to rate the quality of evidence on each outcome (ZS) [22].

## Statistical analysis

The statistical analysis was performed using Stata 15 SE (Stata Corp) by an expert statistician (NG). We calculated pooled relative risks (RRs) with 95% confidence intervals (CIs) for POD and POCD [23]. The analysis was done according to the timing of neuropsychological test measurements (that is, POD at 1, 2 and 5 days and POCD at 1 and 12 weeks). We only performed statistical analysis if at least two RCTs per group were available. Since the settings of the studies do not match exactly, we applied the random effect model with the DerSimonian–Laird estimation [24]. $I^2$ and $chi^2$ tests were used to quantify statistical heterogeneity and obtain $P$-values, respectively; $P < 0.100$ indicated a significant heterogeneity [24]. To evaluate the effect of the individual studies on the pooled estimate, sensitivity analysis was conducted by omitting studies one by one from the analysis if at least three studies were available per analysis. Trial sequential analysis (TSA) was used to quantify the statistical reliability of data if the condition of the tests were met [25].

Since the number of studies included in the analysis was low, publication bias could not be checked either using graphical tools (e.g. funnel plots) or tests (e.g. Egger's test).

### Study protocol and protocol deviations

The protocol for this study was registered in PROSPERO a priori under registration number CRD42018092981, protocol deviations are described in S1 Appendix.

## Results

### Identification and characteristics of the studies

Fig 1 shows the flow chart of our meta-analysis. In total, we identified 1653 records through database searches and one record through other sources; 1408 of which were screened for eligibility after removing duplicates. A total of 1386 ineligible studies were eliminated after browsing titles. Twenty-two studies were removed based on abstract screening. Finally, fourteen studies were included in the systematic review [26–39] eight of which [29–33, 36–38] were qualified to be in the meta-analysis (for reasons, see Fig 1).

The characteristics of the studies included and those excluded on full-text assessment are summarised in Table 1 [26–39] and S1 Table [40–47], respectively.

Conclusions from studies included in the systematic review are summarised in Table 2 [26–39] with detailed results presented in S2 and S3 Tables.

The Summary of findings table provides a brief synopsis of the analyses (Table 3).

### RoB

Although selective reporting was scarce, most items did not meet the criteria for low RoB; these include random sequence generation, allocation concealment and blinding. Three out of fourteen studies were considered to be at high risk in terms of incomplete data reporting. RoB of the included RCTs is summarised in Figs 2 and 3.

### BIS vs. no BIS: POD

We included three [32, 33, 37] studies in the meta-analysis on POD (Fig 4). Based on pooled data from 2138 cases, the use of BIS did prevent POD 1 day after surgery (16.1% vs. 22.8% for BIS vs. no BIS groups, respectively; RR = 0.71, CI: 0.59 to 0.85 for BIS vs. no BIS comparison), without significant between-study heterogeneity ($I^2$ = 0.0%, $P$ = 0.590) [32, 33, 37].

Two studies [28, 37] based on data from 121 patients showed that the use of BIS did seem to prevent POD within 1–5 days after surgery. In contrast, findings from two studies [26, 34] involving 135 patients showed a neutral effect of the use of BIS monitoring.

### BIS vs. no BIS: POCD

We included three [31–33] studies in the meta-analysis on POCD (Fig 5). Based on pooled data from 1985 cases, the use of BIS did not prevent POCD 1 week after surgery (15.8% vs. 18.8% for BIS vs. no BIS groups, respectively; RR = 0.84, CI: 0.66 to 1.08 for BIS vs. no BIS comparison), without significant between-study heterogeneity ($I^2$ = 25.8%, $P$ = 0.260) [31–33]. The neutral association calculated from the data from 1985 cases proved to be underpowered (indicated by TSA) and therefore insufficient to draw a final conclusion (Fig 6). Based on the pooled data from 2047 cases, the use of BIS did prevent POCD 12 weeks after surgery (6.4% vs. 9.1% for BIS vs. no BIS groups, respectively; RR = 0.71, CI: 0.53 to 0.96 for BIS vs. no BIS comparison), again, without significant between-study heterogeneity ($I^2$ = 0.0%, $P$ = 0.969) [31–33]. Only one study [31] involving 60 patients in groups reported data of POCD 52 weeks after surgery (3.7% vs. 12.5% for BIS vs. no BIS groups, respectively; $P$ = 0.36), indicating no benefit of the intervention.

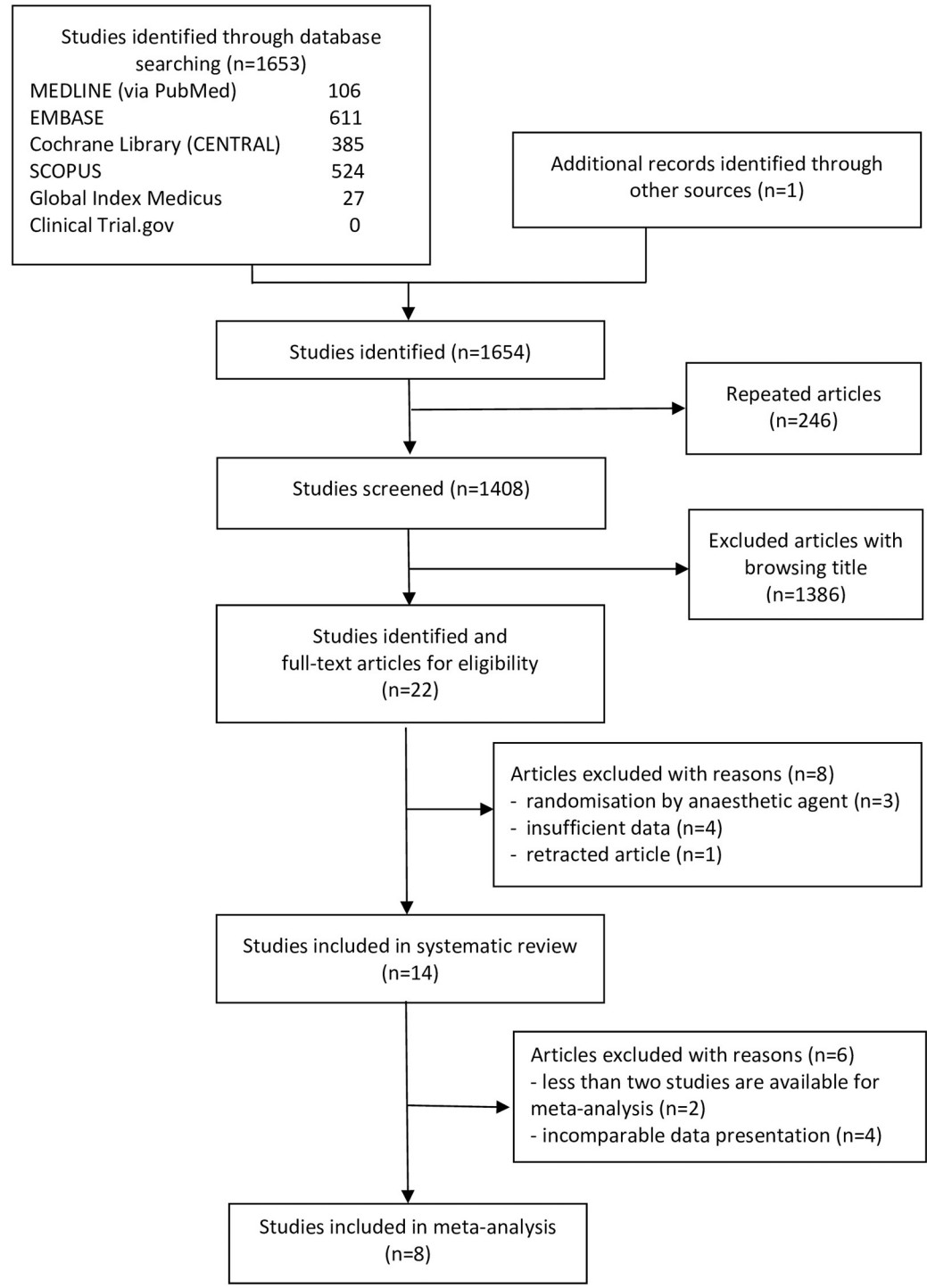

**Fig 1. Flow chart of study selection.**

Two studies [31, 32] with multiple cognitive tests and 154 cases reported a preventive effect of BIS POCD at 1, 12 and 52 weeks.

S2 Table summarises the results of studies reporting on the BIS vs. no BIS comparison for POD and POCD.

**Table 1. Characteristics of the studies included.**

| Author (year, country) | Recruitment period | Study design | Population, Intervention and Comparator | Cognitive test | Outcome, follow-up |
|---|---|---|---|---|---|
| Wong et al. (2002, Canada) [26] | Not stated | RCT | Elderly (≥65 years) with isoflurane-fentanyl anaesthesia undergoing elective orthopaedic surgery, BIS-guided (50–60) (n = 34) vs. no BIS-guided (n = 34) | MMSE, TDT, DSST | POD (at 1–3 days) |
| Farag et al. (2006, USA) [27] | Not stated | RCT | Adults (≥50 years) with isoflurane-fentanyl anaesthesia undergoing spine, abdominal, and pelvic surgery, low BIS (30–40) (n = 36) vs. high BIS (50–60) (n = 38) | MMSE, PSI, WMI, VMI | POCD (at 4–6 weeks) |
| Sadek et al. (2010, Egypt) [28] | Not stated | RCT | Elderly (≥60 years) with desflurane anaesthesia undergoing elective spine surgery, BIS-guided (50–55) (n = 20) vs. MAC-guided (n = 20) | MMSE | POD (at 1–3 days) |
| Sieber et al. (2010, USA) [29] | 2005–2008 | RCT | Elderly (≥65 years) with spinal anaesthesia or propofol sedation undergoing hip fracture repair, low BIS (<50) (n = 57) vs. high BIS (>80) (n = 57) | MMSE, CAM | POD (at 1–2 days) |
| An et al.(2011, China)[30] | Not stated | RCT | Adults (28–65 years) with propofol-remifentanil anaesthesia undergoing receive microvascular decompression, low BIS (30–40) (n = 46) vs. high BIS (55–65) (n = 50) | Mental control, Visional rational, PAVL, DSST, TMT (A), DSF, DSB, Pegboard favoured/no favoured hand | POD (at 5 days) |
| Ballard et al. (2012, UK)[31] | 2007–2009 | RCT | Elderly (≥60 years) with general anaesthesia undergoing elective orthopaedic or abdominal surgery, BIS-used (n = 192) vs. no BIS-used (n = 138) | MMSE, VRT, TMT | POCD (at 1, 12, and 52 weeks) |
| Chan et al. (2013, Hong-Kong)[32] | 2007–2009 | RCT | Elderly (≥60 years) with general anaesthesia undergoing elective major surgery, BIS-guided (40–60) (n = 462) vs. no BIS-guided (n = 459) | MMSE, CFQ, VFT,CAVLT, CTT | POD (at 1 day), POCD (at 1 and 12 weeks) |
| Radtke et al. (2013, Germany)[33] | 2009–2010 | RCT | Elderly (≥60 years) with general anaesthesia undergoing general, abdominal, thoracic, vascular, orthopaedic, otorhinolaryngological, oral and maxillofacial, gynaecological, and urologic surgery, BIS-guided (40–60) (n = 575) vs. no BIS-guided (n = 580) | MMSE | POD (at 1 day), POCD (at 1 and 12 weeks) |
| Altun et al. (2015, Turkey) [34] | Not stated | RCT | Adult (18–40 years) women with sevoflurane or desflurane or regional anaesthesia undergoing Caesarean section, BIS-used (sevoflurane / n = 25, desflurane / n = 25) vs. no BIS-used (n = 25) | MMSE, TDT, CDT | POD (at 1 day) |
| Shu et al.(2015, China)[35] | 2012–2014 | RCT | Young and middle-aged women with sevoflurane-remifentanil anaesthesia undergoing gynaecologic laparoscopic operation, low BIS (30–40 and 40–50) (n = 64 and n = 64) vs. high BIS (50–60) (n = 64) | MMSE, TMT | POD (at 1 day) |
| Hou et al.(2018, China)[36] | Not stated | RCT | Elderly (≥60 years) with sevoflurane-fentanyl anaesthesia undergoing elective total knee arthroplasty, low BIS (40–50) (n = 33) vs. high BIS (55–65) (n = 33) | MoCA | POD (at 1 day) |
| Zhou et al. (2018, China) [37] | 2014–2016 | RCT | Elderly (65–75 years) with general anaesthesia undergoing surgery for resection of colon carcinoma, BIS-guided (40–60)(n = 41) vs. no BIS-guided (n = 40) | MMSE, ANT | POD (at 1–5 days) |
| Sieber et al. (2018, USA) [38] | 2011–2016 | RCT | Elderly (≥65 years) with spinal anaesthesia or propofol sedation undergoing hip fracture repair, low BIS (n = 100) vs. high BIS (n = 100) | MMSE | POD (at 1–5 days) |
| Quan et al. (2019, China) [39] | 2014–2016 | RCT | Elderly (≥60 years) with total intravenous anaesthesia undergoing abdominal surgery, low BIS (40–50) (n = 60) vs. high BIS (50–60) (n = 60) | MMSE, CAM, ANT, Mental control, Visional rational, PAVL, DSST, TMT (A), DSF, DSB, Pegboard favoured/no favoured hand | POCD (at 1 and 12 weeks) |

RCT: randomised controlled trial; BIS: bispectral index; MAC: Minimal Alveolar Concentration; MMSE: Mini Mental State Examination; TDT: Trieger Dot Test; DSST: Digit Symbol Substitution Test; PSI: Parenting Stress Inventory; WMI: Working Memory Index; VMI: Visual Motor Integration Test; CAM: Confusion Assessment Method; PAVL: Paired Associate Verbal Learning; TMT: Trail Making Test; DSF: Digit Span Forward; DSB: Digit Span Backward; VRT: Vigilance Reaction Time; CFQ: Cognitive Failure Questionnaire; VFT: Verbal Fluency Test; CAVLT: Chinese Auditory Verbal Learning; CTT: Color Trial Test; CDT: Clock Drawing Test; MoCA: Montreal Cognitive Assessment; ANT: Attention Network Test; POD: postoperative delirium; POCD: postoperative cognitive dysfunction.

**Table 2. Summary of the conclusions of the studies included.**

| Comparison | Outcome | Follow-up | Conclusion regarding the effect of intervention | | |
|---|---|---|---|---|---|
| | | | **Protective** | **Risk** | **Neutral** |
| BIS vs. no BIS | POD | at 1 day | Sadek[28], Chan[32], Radtke[33], Zhou[37] | | Wong[26], Altun[34] |
| | | at 2–3 days | Zhou[37] | | Wong[26], Sadek[28] |
| | | at 5 days | Zhou[37] | | |
| | POCD | at 1 week | Ballard[31], Chan[32], Radtke[33] | | |
| | | at 12 weeks | Ballard[31], Chan[32] | | Radtke[33] |
| | | at 52 weeks | Ballard[31] | | |
| low (<50) BIS vs. high BIS (≥50) level | POD | at 1 day | Shu[35] | Sieber[29], Hou[36] | Sieber[38] |
| | | at 2 days | | Sieber[29] | Sieber[38] |
| | | at 3–4 days | | | Sieber[38] |
| | | at 5 days | An[30] | | Sieber[38] |
| | POCD | at 1 week | Quan[39] | | |
| | | at 4–6 weeks | Farag[27] | | |
| | | at 12 weeks | | | Quan[39] |

BIS: bispectral index; POD: postoperative delirium; POCD: postoperative cognitive dysfunction

**Table 3. Summary of findings table.**

| P: patients who underwent general anaesthesia, I: BIS monitoring, C: no BIS monitoring, O: postoperative cognitive performance (POD and POCD) | | | | | |
|---|---|---|---|---|---|
| **Outcomes** | **Illustrative comparative risk** | | **Relative effect (95% CI)** | **No. of participants (studies)** | **Quality of evidence (GRADE)** |
| | BIS monitoring | no BIS monitoring | | | |
| **POD at 1 day** (raw data) | **16.1 per 100 patients** | **22.8 per 100 patients** | **RR: 0.71 (0.59–0.85)** | 2138 (3) | ●●○○[1] *low* |
| **POCD at 1 week** (raw data) | **15.8 per 100 patients** | **18.8 per 100 patients** | **RR: 0.84 (0.66–1.08)** | 1985 (3) | ●○○○[2] *very low* |
| **POCD at 12 weeks**(raw data) | **6.4 per 100 patients** | **9.1 per 100 patients** | **RR: 0.71 (0.53–0.96)** | 2047 (3) | ●○○○[3] *very low* |
| **POCD at 52 weeks** (raw data) | **3.7 per 100 patients** | **12.5 per 100 patients** | **RR: 0.32 (0.04–2.72)** | 59 (1) | ●○○○[4] *very low* |
| P: patients who underwent general anaesthesia, I: low level of BIS, C: high level of BIS, O: postoperative cognitive performance (POD and POCD) | | | | | |
| **Outcomes** | **Illustrative comparative risk** | | **Relative effect (95% CI)** | **No. of participants (studies)** | **Quality of evidence (GRADE)** |
| | low level of BIS | high level of BIS | | | |
| **POD at 1 day** (raw data) | **26.2 per 100 patients** | **20.1 per 100 patients** | **RR: 1.92 (0.39–9.33)** | 259 (2) | ●○○○[5] *very low* |
| **POCD at 1 week** (raw data) | **19.2 per 100 patients** | **10.3 per 100 patients** | **RR: 0.52 (0.27–1.00)** | 105 (1) | ●○○○[6] *very low* |
| **POCD at 12 weeks** (raw data) | **39.6 per 100 patients** | **14.6 per 100 patients** | **RR: 0.73 (0.22–2.41)** | 83 (1) | ●○○○[7] *very low* |
| **POCD at 52 weeks** | **no data** | **no data** | **no data** | | |

[1]downgraded one level for risk of bias and one level for indirectness;

[2]downgraded two levels for risk of bias and one level for imprecision;

[3]downgraded two levels for risk of bias and one level for indirectness;

[4]dowgraded one level for risk of bias and two levels for imprecision;

[5]downgraded two levels for risk of bias, two levels for imprecision and one level for indirectness;

[6]downgraded one level for risk of bias and two levels for imprecision;

[7]downgraded one level for risk of bias, two levels for imprecision and one level for indirectness.

BIS: bispectral index; CI: confidence interval; RR: relative risk; POD: postoperative delirium; POCD: postoperative cognitive dysfunction.

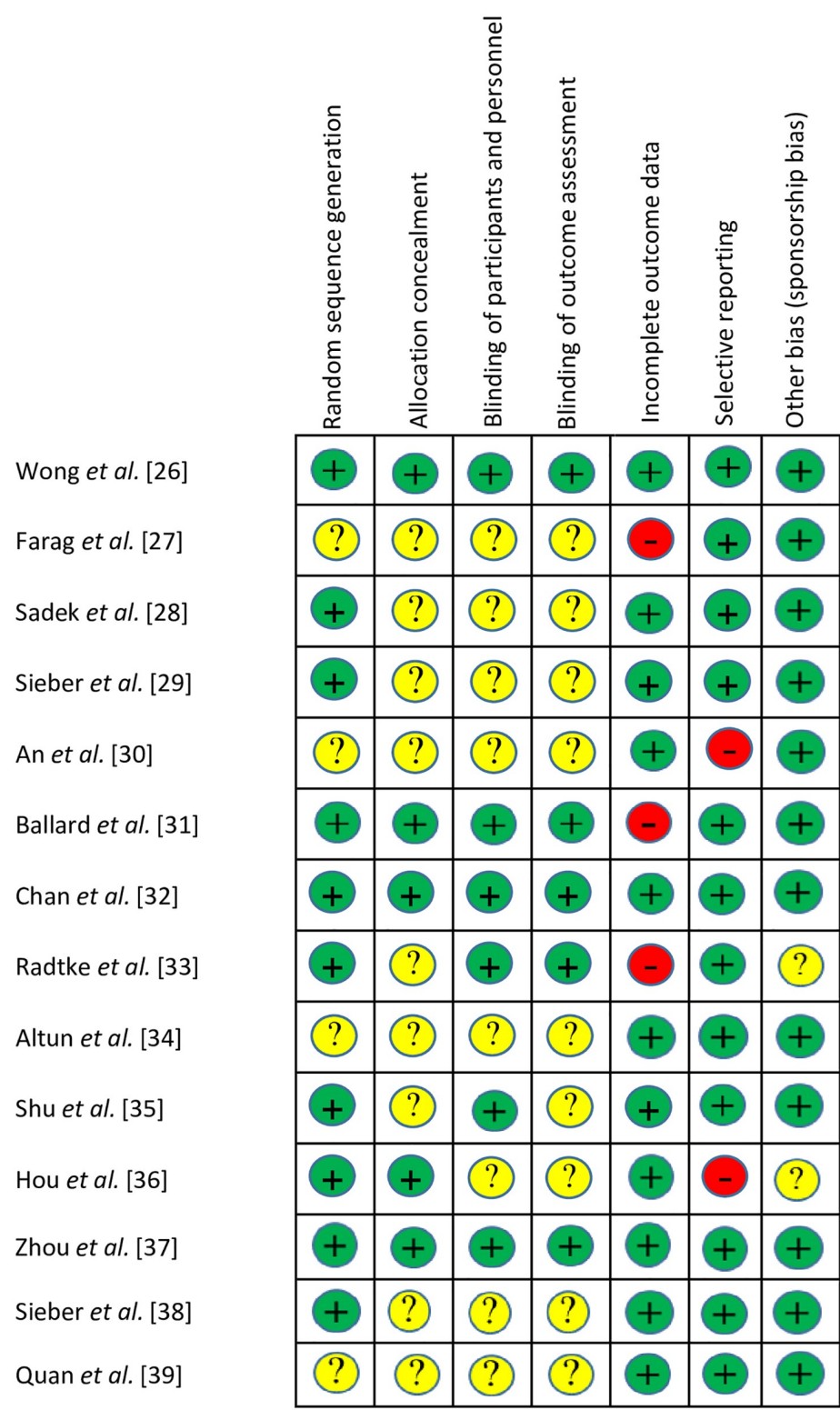

**Fig 2. Risk of bias table.** RCT: randomised controlled trial; "+": low risk of bias; "?": unclear risk of bias; "−": high risk of bias.

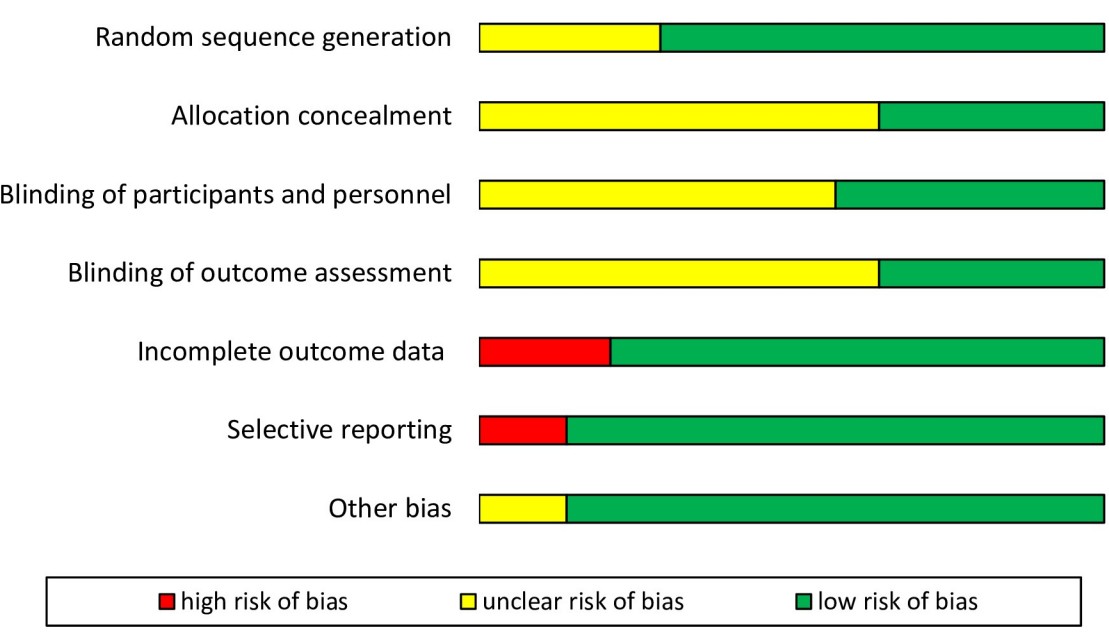

**Fig 3. Risk of bias graph.**

## Low BIS vs. high BIS: POD

We included four [29, 30, 36, 38] studies in the meta-analysis on POD. Data were available in two studies for POD at 1 day [36, 38], POD at 2 days [29, 37] and POD at 5 days [30, 38]. As

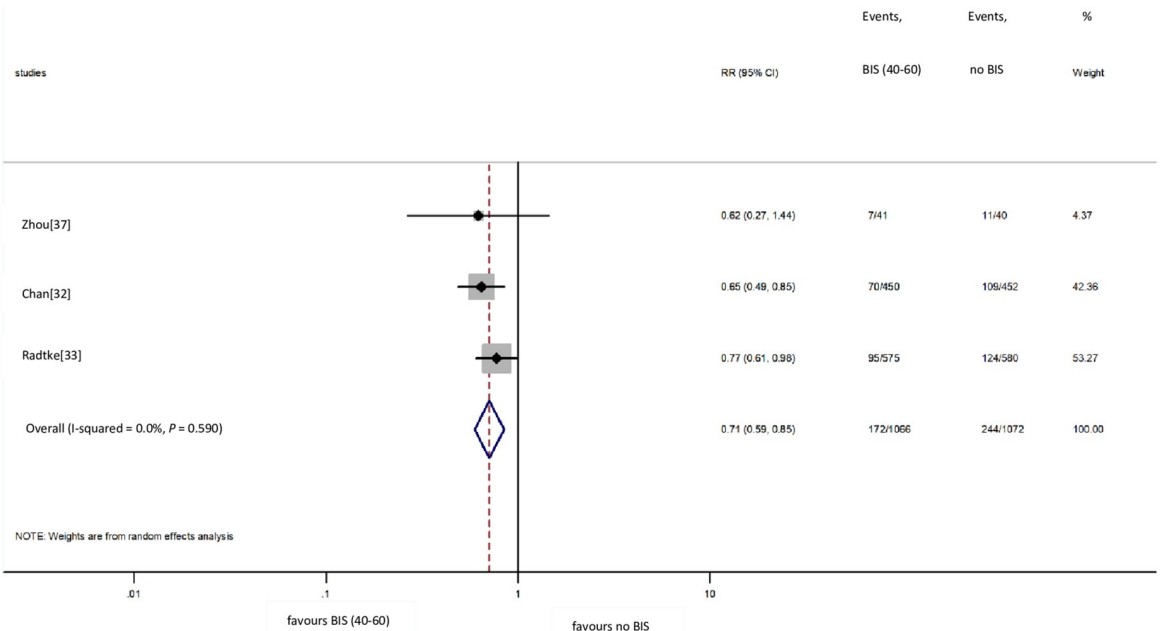

**Fig 4. Risk of POD at 1 day with BIS vs. without BIS monitoring.** POD: postoperative delirium; BIS: bispectral index; RR: relative risk; Cl: confidence interval.

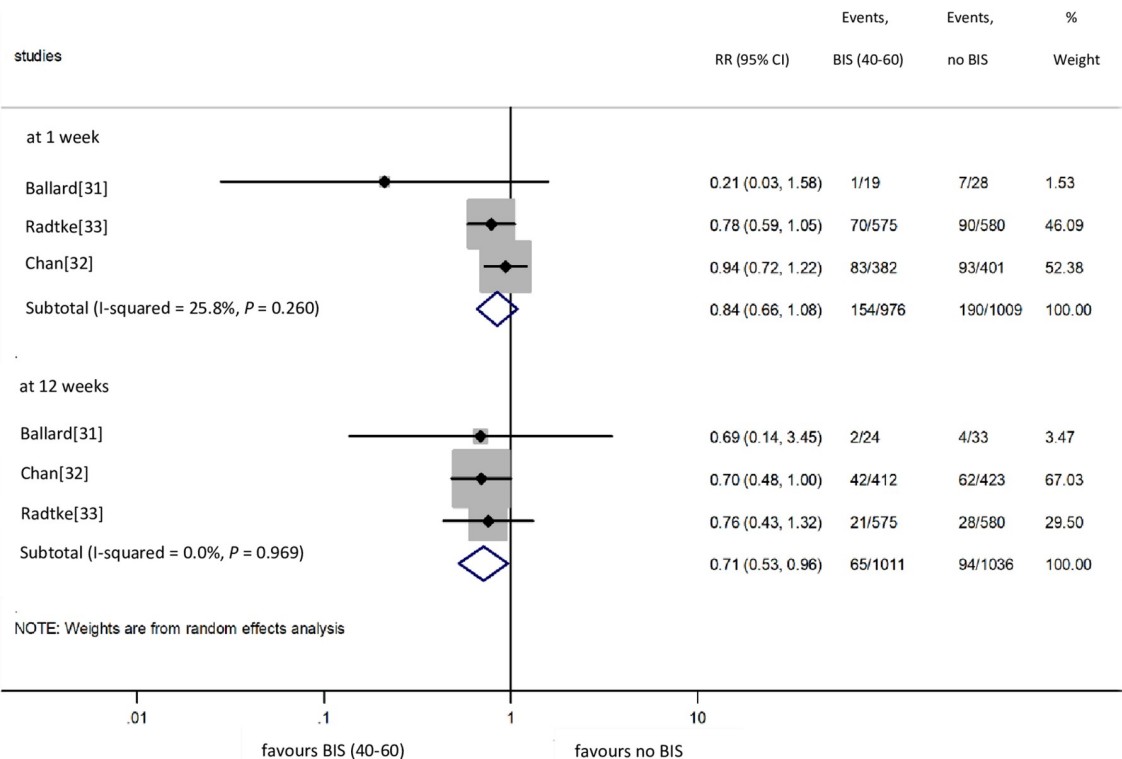

**Fig 5. Risk of POCD at 1 and 12 weeks with BIS vs. without BIS monitoring.** POCD: postoperative cognitive dysfunction; BIS: bispectral index; RR: relative risk; CI: confidence interval.

demonstrated in Fig 7, low BIS did not change the risk of POD at 1 and 5 days, while high BIS proved to be favourable at 2 days (RR = 1.91, CI: 1.13 to 3.22 for the low BIS vs. high BIS comparison), without significant between-study heterogeneity ($I^2$ = 0.0%, $P$ = 0.589).

One study [29] involving 114 patients reported no significant difference in Mini Mental State Examination (MMSE) between groups for POD at 2 days (20.0 ± 9.3 in the low BIS group vs. 23.1 ± 5.5 in the high BIS group; $P$ = 0.08). Another study [35] involving 192 patients attributed a protective effect to low BIS for POD at 1 day $P$ = 0.006 for MMSE scores and $P$ = 0.01 for TMT (Trail Making Test) scores.

### Low BIS vs. high BIS: POCD

Only one study [39] involving 120 patients presented data on POCD at 1 week. Results from groups were significantly different for POCD at 1 week (19.2% in the low BIS group vs. 39.6% in the high BIS group; $P$ = 0.032) and were similar at 12 weeks (10.3% in the low BIS group vs. 14.6% in the high BIS group; $P$ = 0.558).

Two studies [27, 39] involving 154 patients reported similar results on POCD at 1 [27] and 4–6 weeks [39] after surgery. These studies demonstrated a protective effect of lower BIS on POCD. Only one study [39] reported no significant difference between the effects of different BIS levels at 12 weeks.

S3 Table illustrates the results of postoperative cognitive performance tests for the low BIS vs. high BIS comparison.

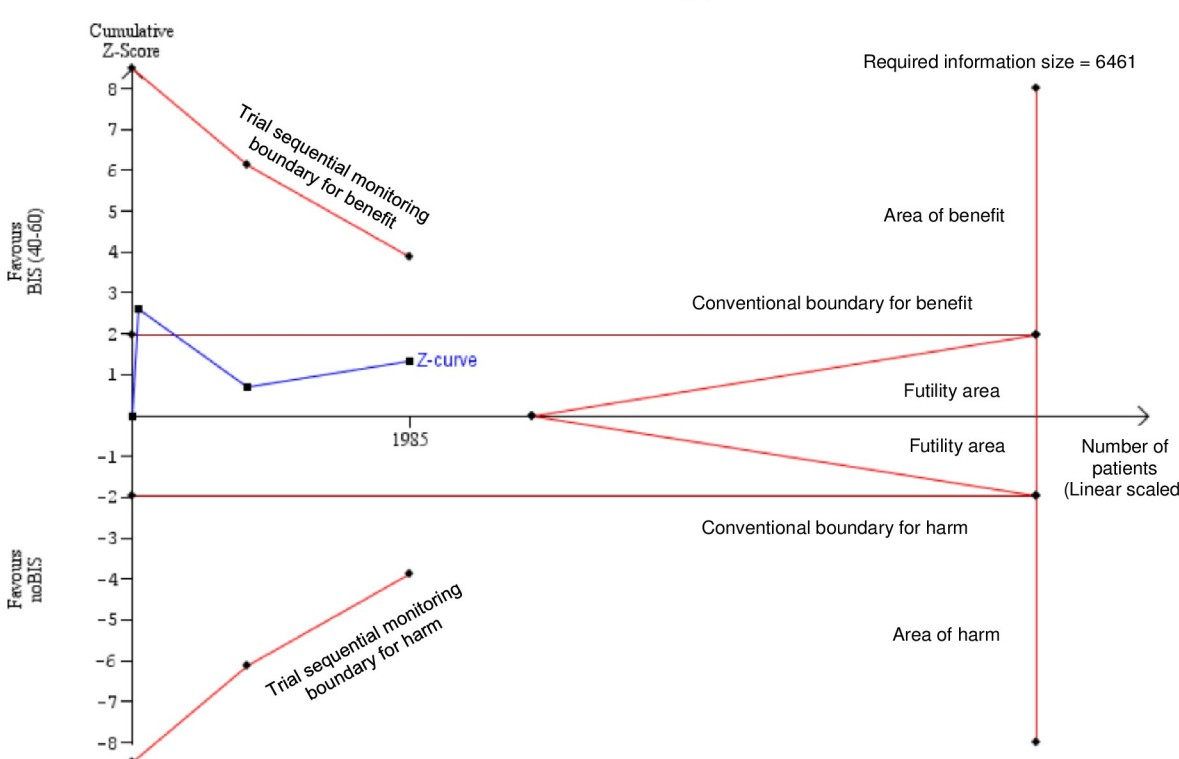

**Fig 6. Trial sequential analysis of data in the BIS vs. no BIS monitoring anaesthesia comparison for the outcome of POCD.** Trial sequential analysis (TSA) is a random effect-based meta-analytical model to estimate the "required information size"; in other words, the required meta-analytical sample size allowing us to draw a confident conclusion. Each dot on the Z-curve represents a new piece of information, the results of a new randomised study (a total of three studies were used in our case). If the Z-curve crosses the futility boundary, the intervention has no significant effect on the outcome and the results are unlikely to change if further studies are added. If the Z-curve, crosses the significance boundaries, the intervention has a significant effect on the outcome. In our case, neither the conventional significance boundary nor trial sequential significance boundary was crossed by the cumulative Z-curve indicating that the meta-analytical sample size (1985 patients) is insufficient to draw a confident conclusion: further studies are needed until the "required information size" (6461 patients) is reached. BIS: bispectral index; POCD: postoperative cognitive dysfunction.

### Sensitivity analysis

Results for POD at 1 day and POCD at 1 week remained unchanged if any studies were removed from the analysis. However, we lost significance if we removed the Chan et al. study from the analysis on POCD at 12 weeks (probably due to the lack of statistical power).

### Discussion

Appropriate brain function monitoring (electroencephalogram monitoring and depth of anaesthesia) would be important to ensure personalised, patient-specific anaesthesia. Theoretically, the application of BIS monitoring could reduce the incidence of prolonged recovery and delayed return of normal cognitive abilities (i.e., orientation and other cognitive functions). A quick and safe postoperative cognitive recovery, such as the avoidance of POD, is of critical importance for patient safety, reduction, and prevention of postoperative complications, early mobilisation and discharge and cost-effectiveness of surgery [2, 6, 7, 9]. POCD starts from the end of the first postoperative week and may persist for weeks to months [2, 11]. POCD impairs quality of life and reduces the Quality-Adjusted life-year (QALY) [11].

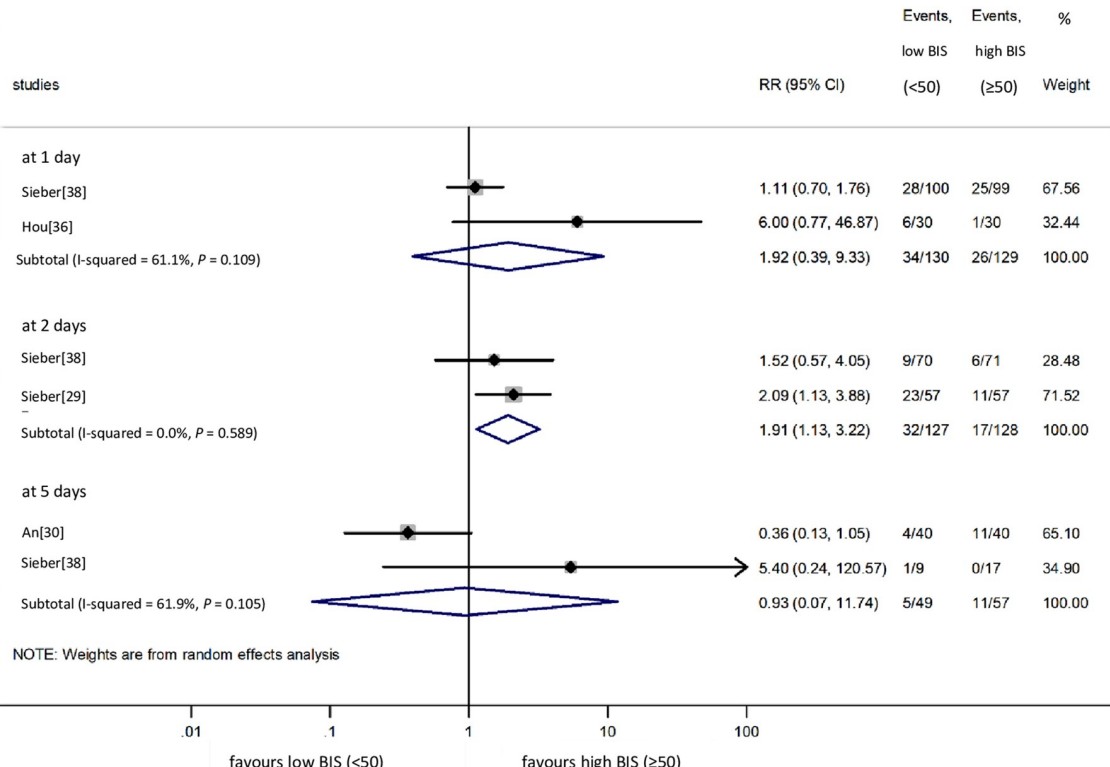

**Fig 7. Risk of POD at 1, 2 and 5 days with low BIS vs. high BIS monitoring.** POD: postoperative delirium; BIS: bispectral index; RR: relative risk; Cl: confidence interval.

Several trials have been conducted to date to evaluate the benefit of BIS monitoring under anaesthesia to prevent POD and POCD (Table 2); however, they have yielded discrepant conclusions (Table 3). Our aim was to summarise the true benefits of BIS monitoring by conducting a meta-analysis with a systematic review.

Our findings provide low quality of evidence that the use of BIS monitoring was superior to not using BIS for POD at 1 day and for POCD at 12 weeks (*very low* quality of evidence). However, the beneficial effects remained undetected for POCD at 1 week, although the analysis was underpowered (see TSA in Fig 6). (*very low* quality of evidence) (Table 3). In addition, low BIS seems to be protective against POD at 2 days and maybe against POCD at 1 week (very low grade of evidence for both) (Fig 7, Table 3).

Three previous meta-analyses [15–17] and two systematic reviews [18, 19] have investigated the association between the depth of anaesthesia and cognitive impairment. Lu et al. [15] compared low BIS and high BIS groups in four studies and concluded that the depth of anaesthesia did not correlate with the risk of POCD, but deep anaesthesia carried a significantly increased risk of POD. In their meta-analysis, the number of eligible studies was relatively low and the merged results on the outcomes were inconsistent. Oliveira et al. [16] demonstrated that BIS monitoring is favourable for POD and POCD at 1 month, as opposed to POCD at 1 week. However, in the latter case, data collection might be compromised since neither event numbers nor total number of patients included in analyses match that reported by the original Chan et al. article for POCD [30]. When pooling dichotomous and continuous outcomes, Mackenzie et al. [17] found that electroencephalogram-guided anaesthesia was associated with a reduction in POD incidence. Luo and Zou reported similar results as Mackenzie [17] on the association of

BIS- and AEP-controlled anaesthesia with POD. Furthermore, a significant association was found with regard to the reduction of long-term cognitive decline. They identified significant heterogeneity across studies but ORs were similar for cardiac and no-cardiac surgeries [18]. Orena et al. concluded based on their research and currently available data that the use of intraoperative anaesthesia depth monitor is recommended during lighter sedation. Furthermore, the prudent use of premedications was also highlighted [19]. Contrasting these studies, we separated dichotomous and continuous data in the analysis, complemented the list of included studies with new ones and used TSA to decide whether not observing a difference between groups can be attributed to the size of the sample (beta-type error) or to a true association.

### Strengths and limitations

(1) This is the first comprehensive assessment to discuss the effect of BIS monitoring on both POD and POCD with rigorous evidence synthesis (as seen in Table 3). In our study, data were shorted to the time of sampling. (2) The main strength of this study is that we included RCTs exclusively [26, 39]. (3) We conducted a comprehensive search with rigorous selection and RoB assessment (Figs 2 and 3). (4) Our main comparisons included a homogenous data set; therefore, confounding factors are unlikely to bias our results (see the $I^2$ and $chi^2$ tests results). This contrasts with the fact that the measurement of cognitive performance is based due to the lack of uniform, comprehensive and ecologically valid tests.

Besides the strengths, the evidence acquired is limited for a number of reasons (Table 3). (1) Although there are many publications, study protocols and data reporting are discrepant and incomplete (e.g. tests handled as continuous variables), thereby impeding statistical analysis. (2) There were mild differences in the definition of POD and POCD across studies, especially in the execution of postoperative cognitive measurements, although this discrepancy did not cause statistical heterogeneity in most analyses. (3) Publication bias could not be assessed due to the low number of eligible studies included. (4) For the same reason, subgroup analyses would be inconclusive. (5) Statistical heterogeneity occurred in some analyses, a result which might be explained by clinical heterogeneity (e.g. indications and types of surgery) and methodological heterogeneity (e.g. perioperative medications). However, previous studies [42, 43, 47] have suggested that certain intraoperative anaesthetic agents (e.g. propofol or volatile gases) may not affect POD and POCD; therefore, they are unlikely to distort our results. (6) Despite the high number of patients included, TSA on the BIS vs. no BIS comparison for POCD at 1 week indicated that neither the conventional significance boundary nor the trial sequential significance boundary was crossed by the cumulative Z-curve (Fig 6). The required sample size would thus be 6461 patients to draw a final conclusion, whereas our meta-analysis of three RCTs included only 1985 cases. (7) It is possible that the neutral association identified in the comparison of low BIS vs. high BIS regarding POD at 1 day is the consequence of beta-type error (Fig 7), which, unfortunately, could not be tested because the conditions of TSA were not met in this case.

### Conclusion

BIS monitoring might have a protective effect against POD at 1 day and POCD at 12 weeks compared to not using BIS while low BIS seems to be favourable regarding the incidence of POD at 2 days and POCD at 1 week. Considering the grade of evidence, these findings call for further investigation to identify those patients by age, comorbid conditions, mental status, type of surgery and anaesthesia and by other individual variables who would benefit the most from the use of BIS or other entropy monitoring as well as to establish the optimal BIS value during anaesthesia.

## Supporting information

**S1 Table. Characteristics of the studies excluded.**
(DOCX)

**S2 Table. Parameters for the BIS vs. no BIS comparison for the results of postoperative cognitive performance to establish the diagnosis of POD and POCD.**
(XLSX)

**S3 Table. Parameters for the low BIS vs. high BIS comparison for the results of postoperative cognitive performance to establish the diagnosis of POD and POCD.**
(XLSX)

**S1 Appendix. Methodological details.**
(DOCX)

**S1 Checklist. PRISMA checklist.**
(DOC)

## Acknowledgments

The authors would like to express their gratitude to anaesthesiologists Professor Lajos Bogar and Professor Zsolt Molnar for the constructive criticism expressed about the manuscript, which significantly improved the paper.

## Author Contributions

**Conceptualization:** Timea Bocskai.

**Data curation:** Timea Bocskai, Márton Kovács, Kázmér Karádi.

**Formal analysis:** Zsolt Szakács, Noémi Gede, Péter Hegyi, Gábor Varga, István Pap, István Tóth.

**Investigation:** Zsolt Szakács.

**Methodology:** Timea Bocskai, Zsolt Szakács, Péter Hegyi, Gábor Varga, István Pap, István Tóth, Kázmér Karádi.

**Project administration:** Márton Kovács.

**Software:** Márton Kovács, Zsolt Szakács, Noémi Gede.

**Supervision:** Zsolt Szakács, Péter Hegyi, Gábor Varga, István Pap, István Tóth, Péter Révész, István Szanyi, Adrienne Németh, Imre Gerlinger, Kázmér Karádi, László Lujber.

**Visualization:** Timea Bocskai, Márton Kovács, Zsolt Szakács, Péter Révész, István Szanyi, Adrienne Németh, Imre Gerlinger.

**Writing – original draft:** Timea Bocskai, Zsolt Szakács, László Lujber.

**Writing – review & editing:** László Lujber.

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
