## [Decision Letter · Decision Letter 0]

11 Dec 2019

PONE-D-19-29094

Is the bispectral index monitoring protective against postoperative cognitive decline? A systematic review with meta-analysis

PLOS ONE

Dear Dr.László Lujber

Thank you for submitting your manuscript to PLOS ONE. After careful consideration, we feel that it has merit but does not fully meet PLOS ONE’s publication criteria as it currently stands. Therefore, we invite you to submit a revised version of the manuscript that addresses the points raised during the review process.

I would appreciate if you pay a careful attention to the reviewers' comments in your reply.   

We would appreciate receiving your revised manuscript by Jan 25 2020 11:59PM. To enhance the reproducibility of your results, we recommend that if applicable you deposit your laboratory protocols in protocols.io, where a protocol can be assigned its own identifier (DOI) such that it can be cited independently in the future. For instructions see: http://journals.plos.org/plosone/s/submission-guidelines#loc-laboratory-protocols

We look forward to receiving your revised manuscript.

Kind regards,

Ehab Farag, MD FRCA FASA

Academic Editor

PLOS ONE

Journal Requirements:

2. We noticed this systematic review may have potential overlap with your study:

https://www.ncbi.nlm.nih.gov/pmc/articles/PMC6166333/

Please consider introducing it in the References and adequately discussing it in the present submission.

'This study was supported by an Economic Development and Innovation Operative Program Grant (GINOP 2.3.2-15-2016-00048) and an Institutional Developments for Enhancing Intelligent Specialization Grant (EFOP-3.6.2-16-2017-0006) from the National Research, Development and Innovation Office.'

'The author(s) received no specific funding for this work.'

Please provide an amended Funding Statement that declares *all* the funding or sources of support received during this specific study (whether external or internal to your organization) as detailed online in our guide for authors at http://journals.plos.org/plosone/s/submit-nowPlease state what role the funders took in the study.  If any authors received a salary from any of your funders, please state which authors and which funder. If the funders had no role, please state: "The funders had no role in study design, data collection and analysis, decision to publish, or preparation of the manuscript."

Reviewers' comments:

Reviewer's Responses to Questions

**Comments to the Author**

1. Is the manuscript technically sound, and do the data support the conclusions?

Reviewer #1: Partly

Reviewer #2: Partly

2. Has the statistical analysis been performed appropriately and rigorously? 

Reviewer #1: Yes

Reviewer #2: Yes

3. Have the authors made all data underlying the findings in their manuscript fully available?

Reviewer #1: Yes

Reviewer #2: Yes

4. Is the manuscript presented in an intelligible fashion and written in standard English?

Reviewer #1: Yes

Reviewer #2: Yes

5. Review Comments to the Author

Reviewer #1: Thank you for submitting the article. It combined information form multiple studies according to the PRISMA Statement and had an updated information since previous metanalysis . There were some issues I wanted to clarify/change:

Abstract.

Ln 47 – Results: We included fourteen studies in the systematic review, eight of which were eligible 48 for meta-analysis. Explain why 6 of them were not included

Ln 49: add the word ’groups’, after “no BIS’

Ln 91-93: consider clarifying abbreviations, reconstruct the sentence and clarify: BIS vs no BIS, then patients with BIS were divided in two groups…

Ln 145: why P<0.1 was chosen?

Ln 166-168: Specify in general terms why 6 studies were not included in metanalysis

Ln 171: Table2 is the same as Table S1. Also you should mention Table 3 Please revise table numbers and comments.

Ln 196: What did other contrasted studies, which included 135 patients, show?

Ln 227-228: where results significantly different with P=0.032, and if they were different, why it was not mentioned in the conclusion?

Reviewer #2: The authors are to be commended for a rigorous attempt to answer 2 questions...does BIS monitoring affect the development of POD or POCD and if the occurrence of a high or low BIS number affects the development of these complications. In an analysis of 1985 patients the conclusions drawn suggest that BIS monitoring is protective of POD on day 1 and protective of POCD at week 12. The level of BIS, high or low did not make a difference in the anlaysis. However, I do not believe that these conclusions can be drawn based on several factors that are not identified; age must be considered, type of anesthetic used, duration of anesthesia, type of surgery preoperative condition, preemptive measures against POD, postoperative complications such as pneumonia, infection or fever, and importantly, what if any measures were taken based on BIS monitoring. For example, if the BIS was low, was the degree of sedation reduced?

Several other studies have looked at BIS monitoring. Orena et al concluded that:

“Use of a depth of anesthesia monitor and lighter sedation had the strongest evidence in reduction of POD. Perioperative dexmedetomidine, ketamine, dexamethasone, and antipsychotic administration may reduce the risk of POD” .(Orena EF et al The role of anesthesia in the prevention of postoperative delirium: a systematic review Minerva Anestesiol. 2016;82(6):669-83.) Other measures were probably also effective

Choi et al developed a screening tool that “successfully identified patients at a high risk of POD at admission. The POD prevention project was feasible to implement, effective in preventing delirium, and improved knowledge regarding delirium among the medical staff” (Choi et al . Impact of a delirium prevention project among older hospitalized patients who underwent orthopedic surgery: a retrospective cohort study BMC Geriatr. 2019 Oct 26;19(1):289. doi: 10.1186/s12877-019-1303-z. The emphasis here was on geriatric patients.

Lee et al also noted that POD is a risk factor for later dementia and following a metanalysis concluded that, “POD after hip surgery is a risk factor for incident dementia. Early identification of cognitive function should be needed after surgery and appropriate prevention and treatment for dementia will be required, especially in cases with POD.” Lee SJ, et al Postoperative delirium after hip surgery is a potential risk factor for incident dementia: A systematic review and meta-analysis of prospective studies. Arch Gerontol Geriatr. 2019 Nov 11;87:103977. doi: 10.1016/j.archger.2019.103977. [Epub ahead of print]. Preoperative recognition of dementia should be made and appropriate measures taken such as medication adjustment.

Specific to this manuscript are the following:

1. All the requirements of PLOS One have been met with the exception of the conclusions drawn

2. The paper could be shortened by eliminating explanations of studies not considered relevant. In general it could be simplified.

3. Any reasons suggested for why neither low nor high BIS values impacted POD at day 1 (In contradiction to other studies)?

4. Lines 91-93 are confusing: l1 vs C1…is that the same as I1 vs Chi1 ?

5. Please reconcile the statements “narcotics do not affect POD and POCD” and “deep anesthesia may”?

6. Of the 14 authors, only one appears to be an anesthesiologist. Given that anesthesia has been implicated in POD and POCD, more impute should be provided from this specialty.

7. The authors note that 6461 patients would be required for a final conclusion and given that only 1/3 are included in this analysis, then even a suggested association with BIS monitoring and POD or POCD may not be warranted. Certainly, more detailed and larger studies looking at many more factors are necessary. Given that some decades ago something akin to mass hysteria arose insisting that BIS monitoring should be the standard of care. Medico legal implications resulted. Subsequent studies have disapproved this insistence. Until more is understood about the development especially of POCD in the elderly, universal BIS monitoring should not be even considered as essential. It may be a tool, but there are too many unanswered questions. And there are other entropy monitors.

.

6. PLOS authors have the option to publish the peer review history of their article (what does this mean?). If published, this will include your full peer review and any attached files.

Reviewer #1: No

Reviewer #2: Yes: elizabeth frost

---

## [Author Response · Author response to Decision Letter 0]

25 Jan 2020

Ehab Farag MD FRCA FASA 22 January 2020

Academic Editor

PLOS ONE

Enclosed please find our revised manuscript entitled “Bispectral index monitoring under general anaesthesia is protective against postoperative delirium and postoperative cognitive dysfunction: A systematic review with meta-analysis”. We greatly appreciate the complimentary comments and suggestions made by You and the Reviewers and thank you for being thorough and for providing us with detailed feedback. We have amended the manuscript as per the recommendations. Based on the reviewers comments we rearranged the sequence of some content. All comments by You and the reviewers were addressed point-by-point to our best knowledge.

As recommended, we removed funding sources from the paper and discussed (and cited) the relevant summary publications mentioned in the letter.

The authors of this manuscript are: Tímea Bocskai, Márton Kovács, Zsolt Szakács, Noémi Gede, Péter Hegyi, Gábor Varga, István Pap, István Tóth, Péter Révész, István Szanyi, Adrienne Németh, Imre Gerlinger, Kázmér Karádi, László Lujber. All authors have reviewed and approved the final revised version of the manuscript.

Registration number: PONE-D-19-29094.

We hope that our revised manuscript meets the high standards of the Journal and that it is now acceptable for publication.

Thank you for giving us this opportunity. We look forward to hearing from you.

Sincerely,

Tímea Bocskai MD PhD

Response to the Reviewers’ comments

Changes are highlighted with the ‘track changes’ function of Microsoft Word.

(Revised Manuscript with Track Changes)

Responses to Reviewer #1

Reviewer #1 Thank you for submitting the article. It combined information form multiple studies according to the PRISMA Statement and had an updated information since previous meta-analysis. There were some issues I wanted to clarify/change:

[Comments from the authors] We would like to thank Reviewer #1 for his/her excellent comments, which have significantly improved the quality of our manuscript.

Abstract

Question Nº 1: Ln 47: Results: We included fourteen studies in the systematic review, eight of which were eligible 48 for meta-analysis. Explain why 6 of them were not included.

[Reply] Thank you for the comment. We added the reasons for exclusion to Fig 1.

[Change in the manuscript] Fig 1

Question Nº 2: Ln 49: add the word ’groups’, after “no BIS’

[Reply] Done.

[Change in the manuscript] Abstract Ln 54

Material and methods

Question Nº 1: Ln 91-93: consider clarifying abbreviations, reconstruct the sentence and clarify: BIS vs no BIS, then patients with BIS were divided in two groups…

[Reply] Thank you for comment. All related abbreviations are clarified in the introduction section, and, to ease the understanding, we spelt out the elements of the PICO framework in this section. In addition, we re-arranged the position of these elements within the paragraph; therefore, PICO elements directly precede the corresponding information in the current version of the manuscript.

[Change in the manuscript] Material and methods Ln 98-102

Question Nº 2: Ln 145: why P<0.1 was chosen

[Reply] Thank you for this comment. The cut-off was chosen as per the recommendations of the Cochrane Handbook for Systematic Reviews of Interventions Version 5.1.0. 2011. The cut-off deviates from the usual 0.05 to reduce the chance of beta-type error.

https://handbook-5-1.cochrane.org/chapter_9/9_5_2_identifying_and_measuring_heterogeneity.htm

[Change in the manuscript] References [21]

Results

Question Nº 1: Ln 166-168: Specify in general terms why 6 studies were not included in metanalysis.

[Reply] Thank you for the comment. We added the reasons for exclusion to the Fig 1.

[Change in the manuscript] Results Ln 178 and Fig 1

Question Nº 2: Ln 171: Table2 is the same as Table S1. Also you should mention Table 3 Please revise table numbers and comments.

[Reply] We maximally agree with this comment. Indeed, we wrongly inserted the tables in the manuscript before submission. We re-checked all tables and figures and amended the errors. Please find the corrected tables.

[Change in the manuscript] Results Pg 12-15 (Table 2 and 3) and Supplementary (Table S1)

Question Nº 3: Ln 196: What did other contrasted studies, which included 135 patients, show?

[Reply] They found neutral associations. We added it to the corresponding sentence.

[Change in the manuscript] Results Ln 214-216

Question Nº 4: Ln 227-228: Where results significantly different with P=0.032, and if they were different, why it was not mentioned in the conclusion?

 [Reply] Thank you for the comment. We complemented the discussion and the conclusion and reported this piece of evidence as well.

[Change in the manuscript] Ln 280-281 and Ln 342-344

Responses to Reviewer #2

Reviewer #2: General comment: The authors are to be commended for a rigorous attempt to answer 2 questions...does BIS monitoring affect the development of POD or POCD and if the occurrence of a high or low BIS number affects the development of these complications. In an analysis of 1985 patients the conclusions drawn suggest that BIS monitoring is protective of POD on day 1 and protective of POCD at week 12. The level of BIS, high or low did not make a difference in the anlaysis. However, I do not believe that these conclusions can be drawn based on several factors that are not identified; age must be considered, type of anesthetic used, duration of anesthesia, type of surgery preoperative condition, preemptive measures against POD, postoperative complications such as pneumonia, infection or fever, and importantly, what if any measures were taken based on BIS monitoring. For example, if the BIS was low, was the degree of sedation reduced?

Several other studies have looked at BIS monitoring. Orena et al concluded that:

“Use of a depth of anesthesia monitor and lighter sedation had the strongest evidence in reduction of POD. Perioperative dexmedetomidine, ketamine, dexamethasone, and antipsychotic administration may reduce the risk of POD”. (Orena EF et al The role of anesthesia in the prevention of postoperative delirium: a systematic review Minerva Anestesiol. 2016;82(6):669-83.) Other measures were probably also effective.

Choi et al developed a screening tool that “successfully identified patients at a high risk of POD at admission. The POD prevention project was feasible to implement, effective in preventing delirium, and improved knowledge regarding delirium among the medical staff” (Choi et al . Impact of a delirium prevention project among older hospitalized patients who underwent orthopedic surgery: a retrospective cohort study BMC Geriatr. 2019 Oct 26;19(1):289. doi: 10.1186/s12877-019-1303-z. The emphasis here was on geriatric patients.

Lee et al also noted that POD is a risk factor for later dementia and following a metanalysis concluded that, “POD after hip surgery is a risk factor for incident dementia. Early identification of cognitive function should be needed after surgery and appropriate prevention and treatment for dementia will be required, especially in cases with POD.” Lee SJ, et al Postoperative delirium after hip surgery is a potential risk factor for incident dementia: A systematic review and meta-analysis of prospective studies. Arch Gerontol Geriatr. 2019 Nov 11;87:103977. doi: 10.1016/j.archger.2019.103977. [Epub ahead of print]. Preoperative recognition of dementia should be made and appropriate measures taken such as medication adjustment.

[Comments from the authors] We would like to thank Reviewer #1 for her excellent comments, which have significantly improved the quality of our manuscript. We entirely agree with your comments and strove to modify the conclusion accordingly.

Question Nº 1: All the requirements of PLOS One have been met with the exception of the conclusions drawn

[Reply] Thank you for the comment. We have revised the evidence and reconsidered the conclusions accordingly. Please, indicate if further fine-tuning is required.

[Change in the manuscript] Discussion and Conclusion

Question Nº 2: The paper could be shortened by eliminating explanations of studies not considered relevant. In general it could be simplified.

[Reply] To ease the understanding of the paper, we simplified the methods section by relocating sections not essential for the understanding to the appendix. Only studies meeting our eligibility criteria are detailed in the results section but to allow the readers a quick summary of the process, the findings and grade of evidence are summarized in Tables 2 and 3. Unfortunately, as Reviewer #1 pointed to it correctly, we wrongly inserted the tables in the manuscript, which was amended in the current version. We hope that these changes made the paper easier to read but we are open to further simplifications if needed.

[Change in the manuscript] Material and methods

Question Nº 3: Any reasons suggested for why neither low nor high BIS values impacted POD at day 1 (In contradiction to other studies)?

[Reply] Thank you for the thought-provoking question. We consulted a statistician on this issue and concluded that the neutral association might be the consequence of beta-type error (TSA could not be executed in this case due to technical reasons). We used the random-effect model in the analyses to overcome the differences in the settings of the studies included, which is accompanied by the widening of the confidence intervals (in other words, results are considered less precise to reduce the risk of establishing false positive associations). In addition, the neutral effect detected by Sieber et al. was taken into account in the analysis when calculating the final estimate. We indicated this as a limitation in the manuscript.

[Change in the manuscript] Results Ln 235-240 and Discussion 276-281

Question Nº 4: Lines 91-93 are confusing: l1 vs C1…is that the same as I1 vs Chi1 ?

[Reply] Thank you for this comment. We spelt out the abbreviation to prevent any misunderstanding from occurring.

[Change in the manuscript] Material and Methods Ln 97-100

Question Nº 5: Please reconcile the statements “narcotics do not affect POD and POCD” and “deep anesthesia may”?

[Reply] Thank you for this comment. The wording has been clarified according to the references cited.

[Change in the manuscript] Discussion Ln 325-327

Question Nº 6: Of the 14 authors, only one appears to be an anesthesiologist. Given that anesthesia has been implicated in POD and POCD, more impute should be provided from this specialty.

[Reply] Thank you for this comment. We invited two expert anesthesiologists for reviewing the manuscript and considered their opinion as well when revising the manuscript. Please, find their opinion attached. Unfortunately, in our opinion, this contribution did not meet the ICMJE authorship policy; therefore, we indicated their contribution within the acknowledgements.

[Change in the manuscript] Additional file, Response to Reviewer (opinion of independent reviewers)

Question Nº 7:

[Reply] We maximally agree with this comment and rephrased the conclusion accordingly.

[Change in the manuscript] Discussion and Conclusion

---

## [Editor Report · Decision Letter 1]

29 Jan 2020

Is the bispectral index monitoring protective against postoperative cognitive decline? A systematic review with meta-analysis

PONE-D-19-29094R1

Dear Dr. László Lujber

We are pleased to inform you that your manuscript has been judged scientifically suitable for publication and will be formally accepted for publication once it complies with all outstanding technical requirements.

With kind regards,

Ehab Farag, MD FRCA FASA

Academic Editor

PLOS ONE
---

## [Editor Report · Acceptance letter]

5 Feb 2020

PONE-D-19-29094R1 

Is the bispectral index monitoring protective against postoperative cognitive decline? A systematic review with meta-analysis 

Dear Dr. Lujber:

I am pleased to inform you that your manuscript has been deemed suitable for publication in PLOS ONE. Congratulations! Your manuscript is now with our production department. 

With kind regards,

on behalf of

Dr. Ehab Farag 

Academic Editor

PLOS ONE